# Paleo-Sedimentary Environments and Controlling Factors for Enrichment of Organic Matter in Alkaline Lake Sediments: A Case Study of the Lower Permian Fengcheng Formation in Well F7 at the Western Slope of Mahu Sag, Junggar Basin

**Gangqiang Chen [1], Yuantao Tang [1], Yuhang Nan [1,*], Fan Yang [2] and Dongyong Wang [1]**

[1] Faculty of Petroleum, China University of Petroleum-Beijing at Karamay, Karamay 834000, China; dywang0907@163.com (D.W.)
[2] Research Institute of Petroleum Exploration and Development, PetroChina, Beijing 100083, China
* Correspondence: nyh_0216@163.com

**Abstract:** The Lower Permian Fengcheng formation is a significant source of rocks and a reservoir for the Mahu Sag in the Junggar Basin. Recently, the paleo-environment deposition factors of the $P_1f$ formation have become a popular research topic. This research was conducted using data from the F7 well of Mahu Sag, based on the geochemical analysis results of TOC (total organic carbon), REE (rare earth elements), and major and trace elements of 53 samples from $P_1f$ (Lower Permian Fengcheng formation), and some deposition factors are discussed. The $P_1f$ deposition process was classified into four stages based on paleo-environment elemental indicators. This research describes the deposition process of the evolution of alkaline lakes. The early and preliminary stages of alkali lake evolution are considered as late $P_1f_1$ to middle $P_1f_2$; the paleoclimate of this process was dry, the reduction conditions increased, and the paleo-productivity and lake salinity were enhanced. The terminal stage of alkali lake evolution is considered as late $P_1f_2$ to middle $P_1f_3$; in this period, the paleoclimate changed with seasonal cycles, resulting in a decrease in water salinity and an increase in oxidation; the paleo-productivity of the alkaline lake was at a medium level. Until the end of $P_1f_3$, the salinity of the lake decreased, and the water body became anoxic and weakly alkaline. Furthermore, the research on TOC and sedimentary parameters confirmed that the deposition of $P_1f$ organic matter is affected by multiple types of factors. A relatively warm climate, lack of oxygen, fresh water–brackish water, suitable debris flow, and high primary productivity conditions promoted organic matter deposition.

**Keywords:** depositional environment; organic matter enrichment; major elements; trace elements; Fengcheng formation; Junggar Basin

## 1. Introduction

The Lower Permian Fengcheng formation was deposited in an alkaline lake, and is the oldest high-quality alkaline lacustrine source rock in the world [1]. In this set of source rock, alkaline minerals are rich. The Fengcheng formation source rock is characterized by the development of alkaline minerals, and is rich in bacteria and algae but lacking in eukaryotes in the water body. The bioprecursors of the Fengcheng formation source rocks are dominated by bacteria and algae [1,2], with a high potential for hydrocarbon generation, which may be much larger than that of freshwater lake basins and saltwater lake basins [3,4]. In previous studies, it has also been reported that the alkaline lake was closely related to the development of high-quality source rocks [5,6]. Domagalski et al. (1989) conducted a systematic analysis and comparison of samples from Mono Lake, Great Salt Lake, and Walker Lake, and the results indicated that the TOC values of the alkaline lake sediments were higher than those of ordinary salt lake sediments. By comparing the

samples from alkaline lake sediments and freshwater lake sediments, Horsfield et al. (1994) concluded that the hydrocarbon generation potential of the alkaline lake sediments is five times that of the freshwater lake sediments. Wang and He (2002) conducted a study on the alkaline lacustrine source rocks of the Hetaoyuan Formation of Paleogene in the Biyang Depression of the Nanxiang Basin, and the results suggest that the organic matter content of the alkaline lacustrine source rocks is seven times higher than that of carbonate rocks in eighteen marine basins in the world, and the chloroform bitumen content is two to three times higher than that of those eighteen marine basins. The organic matter of carbonate source rocks deposited in alkaline environments is rich in hydrogen-rich components, with a high hydrocarbon conversion rate [7]. Therefore, these source rocks are good to excellent source rocks. From the perspective of petroleum exploration, the potential source rocks deposited in alkaline lakes are obviously superior to those deposited in freshwater lakes.

The sedimentary environment has a significant influence on organic matter contents [8]. The factors controlling the concentration of organic matter are mainly preservation conditions, production conditions, or the combination of preservation and production conditions [9]. For the alkaline lacustrine sediments of $P_1f$, previous studies mainly focused on the sedimentary environments and depositional mechanisms. However, little study has been conducted on the relationship between organic matter content and sedimentary environments for $P_1f$ on the western slope of Mahu Sag. The mechanisms of the influence of sedimentary environments on the concentration of organic matter are still unclear.

In this paper, the paleo-environments were reconstructed with inorganic geochemical data. The source rocks of alkaline lakes have high economic potential, and an understanding of the genesis, the sedimentary environment, and sedimentary evolution of alkaline lake source rocks is significant. The restoration of the paleo-sedimentary environment is very important for the migration and generation of oil and gas and can be conducted with many methods. The concentrations and ratios of major and trace elements are usually used to analyze the change in paleo-sedimentary environments, and the quantitative analysis of major and trace elements has been conducted widely in many basins [10]. Major and trace elements in sedimentary rock are directly controlled by their physicochemical properties and influenced by the paleoclimates and paleo-sedimentary water bodies, which can provide reliable information on the change in the paleo-environment [11]. In this study, the major and trace elements and their ratios were analyzed with samples that were systematically cored vertically from a single well. Based on the integrated study of the sedimentary environment with multiple indicators, the main controlling factors for the content of organic matter are made clear, which is helpful for the analysis and reconstruction of the time–space sedimentary model of organic matter distribution and hydrocarbon generation of alkaline lacustrine sediments in Mahu Sag.

## 2. Geological Setting

The Junggar Basin is located in the northwestern part of China, as presented in Figure 1, and is one of the six major petroliferous basins in China [12]. In 1955, Karamay Oilfield, the first large oilfield in China, was discovered in the northwest of the Junggar Basin [13]. After 66 years of exploration and development, the proved reserves are distributed over 100 km from the Ke–Bai Fault zone to the Wu–Xia Fault zone in the northwest of the basin. Since 2012, more and more oil and gas discoveries have been made successfully, forming another hundred-kilometer oil province [14]. By 2020, the proved reserves in the Junggar Basin reached 2.6 billion tons of oil equivalent, among which Mahu Sag and its surroundings have the highest concentration of oil and gas resources, with the proved geological reserves in this area accounting for 70% of the whole basin. The hydrocarbons are mainly from the Permian source rock of $P_1f$ in Mahu Sag.

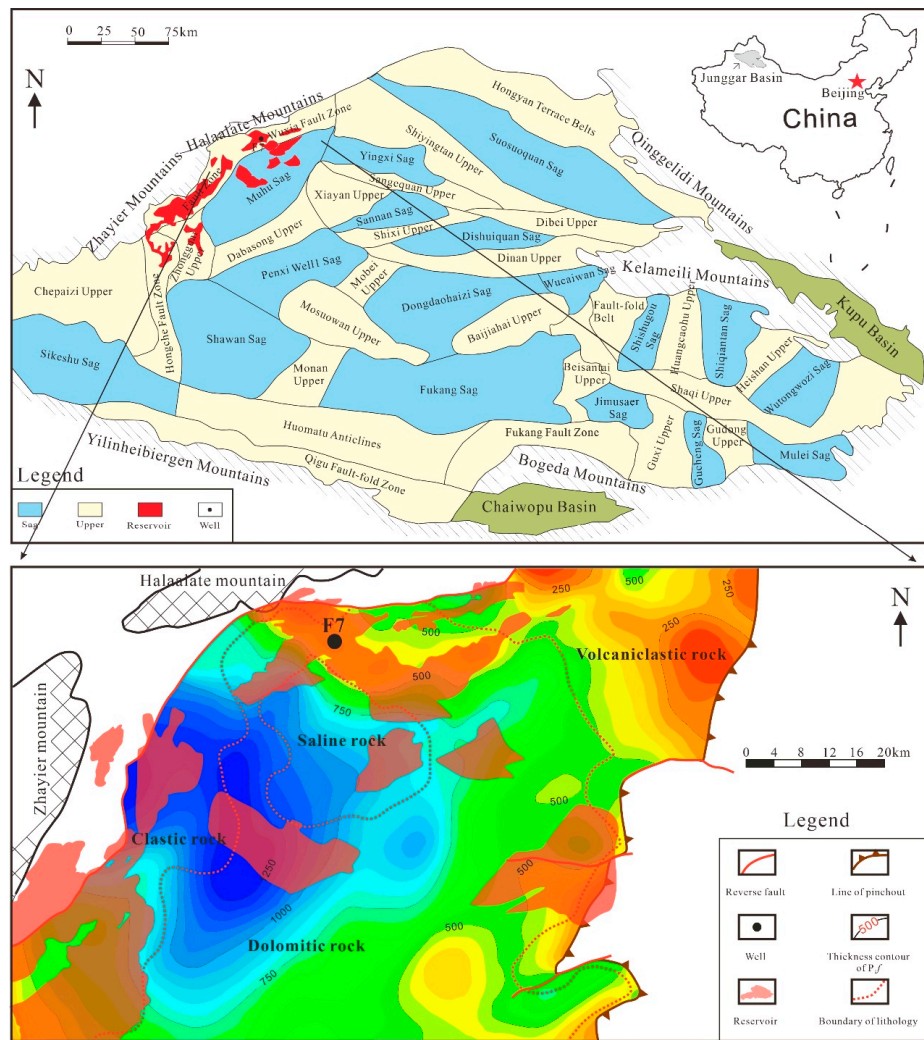

**Figure 1.** Distribution of tectonic units and well locations in Junggar Basin, Northwest China.

The study area is located in the shallow saline lake system, north of the ancient Mahu lake, as shown in Figure 1, near the basin boundary [15]. Prior to the Carboniferous, the marine sedimentary environment was dominant, and the seawater gradually withdrew from the southeast from the Permian, with the shrinking of the ocean southward [16]. The proto-Junggar Basin transformed from an extensional basin to a foreland basin [17]. The $P_1f$ in Mahu Sag was deposited in a typical foreland tectonic setting, with a thickness of 0–1400 m and a burial depth of 3000–6500 m. The Fengcheng formation can be divided into three members in ascending order, including Feng-1 Member ($P_1f_1$), Feng-2 Member ($P_1f_2$) and Feng-3 Member ($P_1f_3$), which thick in the west and thin in the east, forming an eastward-thinning wedge. There are four types of lithology developed in the Fengcheng formation, as shown in Figures 1 and 2. In the northwest margin of Mahu Sag, thick glutenite is mainly developed, with an area of 1720 km$^2$ and a thickness of 500–1000 m. In the central part of Mahu Sag, dolomitic rocks and salt rocks are extensively developed, with an area of 6698 km$^2$ and a thickness generally greater than 500 m. An interbedded area of dolomitic rocks with salt rocks exists in the central part. In the eastern part of Mahu Sag, in addition to glutenite and dolomitic rocks, volcanic clastic rocks are also developed in some intervals, with an area of 1677 km$^2$ and a thickness of 10–34 m. $P_1f$ has a set of high-quality and mature source rocks, with a thickness of 50–400 m and a distribution area of approximately 6000 km$^2$ [18,19].

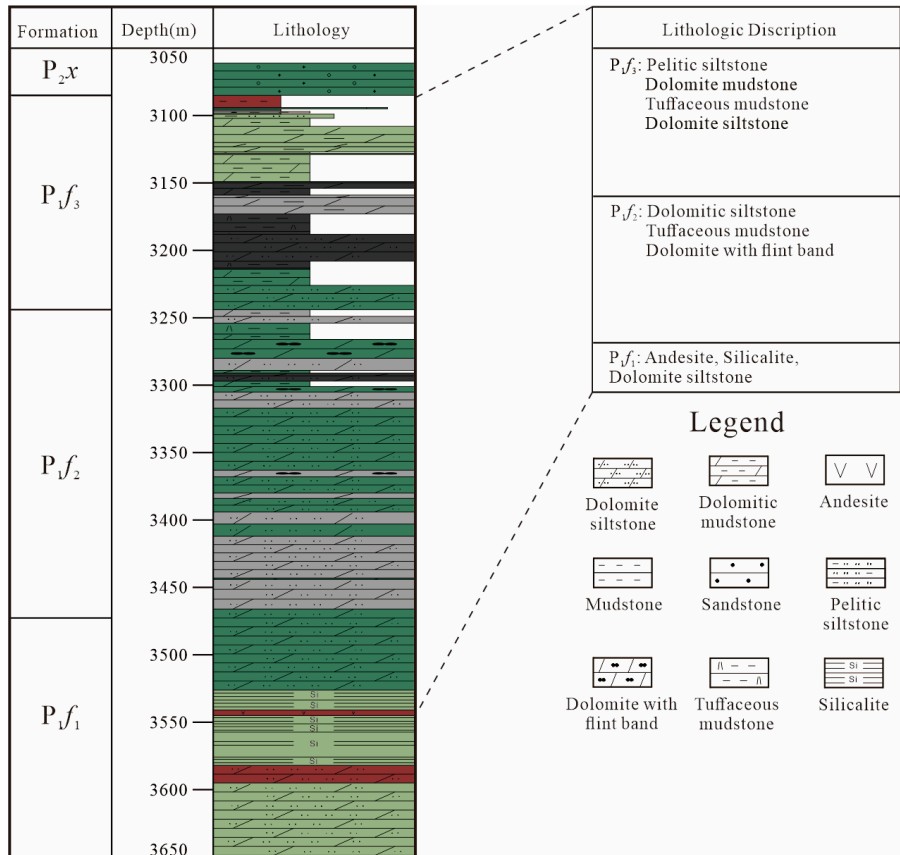

**Figure 2.** Stratigraphic column of Well F7 for the Fengcheng Formation, Mahu Sag.

The burial depth of Fengcheng formation is shallow in the northwest and deep in the southeast. Therefore, due to a large burial depth in the deep basin, the Fengcheng formation has not been penetrated by any wells. In the northwest margin of the Mahu Depression, the Fengcheng formation is shallow in burial depth, was penetrated by many wells [20].

## 3. Sample Collection and Data Processing

### 3.1. Sample Collection

In this study, the samples were collected from the Lower Permian Fengcheng Formation, from Well F7, which is located on the western slope of Mahu Sag and at the northern margin of the ancient Mahu Lake. A total of 53 samples were analyzed for inorganic geochemistry study, of which 5 from $P_1f_1$, 17 from $P_1f_2$ and 31 from $P_1f_3$.

All of the analyses were conducted at Qingdao Sparta Analysis and Testing Co., Ltd. (Qingdao, China). The collected core samples were crushed using a pollution-free crushing machine, then washed, dried, and ground, making the sample powder (<200 mesh) for inorganic geochemistry investigation. The analytical precision of the test results was obtained by using five materials, including the China national standard materials GBW07314, GBW07315, GBW07316, and the international standard materials BHVO-2 and BCR-2, with a test accuracy of over 10%.

For analysis of each element, firstly, 1 g sample was weighed in PTFE vessel, and 1.5 mL HF and 0.5 mL $HNO_3$ were added; the mixture was heated at 150 °C for 12 h. After cooling down, 0.25 mL $HClO_4$ was added, and the resulting solution was heated on a hotplate until dry. Then, 1 ml high-purity water and 1 mL $HNO_3$ were added, heated again at 150 °C for 6 h. Finally, after cooling down, the solution was transferred into a 40 mL polyethylene breaker and diluted to 40 mL with high-purity water after thoroughly mixing.

The contents of major elements (shown as oxides, $Al_2O_3$, MgO, CaO, $Na_2O$, $K_2O$, $P_2O_5$, $TiO_2$, MnO, $Fe_2O_3$) followed the China National Standard GB/T 14506.28-2010 [21], with an accuracy of over 1%. Trace elements and rare earth elements followed China National Standard GB/T 14506.30-2010 [22], and the accuracy was over 1%. The experimental instrument adopts the thermal simulation experiment device for hydrocarbon generation kinetics developed by the Guangzhou Institute of Geochemistry, Chinese Academy of Sciences (patent number: CN101149363). The experimental process and standard follow the oil and gas industry standard gold tube hydrocarbon generation thermal simulation experiment method (SY/T 7035-2016 [23]).

*3.2. Data Processing*

To reconstruct the paleo-environmental conditions with the concentrations of trace elements, we conducted an assessment on whether the elements were relatively enriched or poor using the formula $EF_{element} = (X_{sample}/Al_{sample})/(X_{average\ shale}/Al_{average\ shale})$ [24,25], before the use of element concentrations [26]. This paper used the most common standard used to normalize rare earth elements, the Post Archean Australian Shale (PAAS) values, and yttrium (Y), which is similar in its chemical properties, is included in the rare earth elements (REE) in this paper. Cerium anomaly was calculated using the following equations [27]: $Ce/Ce^{*anomaly} = Ce_N/(La_N \times Pr_N)^{1/2}$.

Samples with a calculated value between 0.9 and 1.1 were considered devoid of anomalies [28]. In addition, $(La/Yb)_N$, $(La/Sm)_N$, and $(Gd/Yb)_N$ values were calculated to reflect the degree of relative fractionation in REE/PAAS patterns. $(La/Yb)_N$ values can describe the degree of fractionation between light REE (LREE) and heavy REE (HREE) together with LREE/HREE, where a large value represents an obvious degree of fractionation. $(La/Sm)_N$ and $(Gd/Yb)_N$ values were used as the parameters of relative fractionation in LREE and HREE, respectively. In this study, LREE includes La, Ce, Pr, Nd, Sm, and Eu; HREE includes Gd, Tb, Dy, Ho, Er, Tm, Yb, Lu, and Y. The REE contents and calculated parameters are shown in Supplementary Table S1. In order to distinguish the main controlling factor affecting the accumulation of organic matter, the partial correlation analyzing method was used to study the correlation between TOC and the influence factors in this study.

**4. Results**

*4.1. Contents of Major and Trace Elements*

The concentrations of major elements, shown as oxides, and trace element contents are listed in Supplementary Tables S2 and S3, respectively. According to experimental results, the most abundant major element oxides in these samples were $SiO_2$ (58.68%), $Al_2O_3$ (9.26%), and CaO (6.02%), with a total average concentration of 73.96%, as shown in Table 1. The next most abundant major element oxides were $Na_2O$, MgO, $K_2O$, and $Fe_2O_3$, while the concentrations of $P_2O_5$, $TiO_2$, MnO were all less than 1%. Figure 3 shows the enrichment or depletion of major elements which are normalized to average shale. Most samples were enriched in Na, Mg, Al, Si, K, Ca, and Fe. For samples from the Fengcheng formation, P and Ti were enriched relative to average shale, while Mn was enriched in $P_1f_1$ and $P_1f_2$, and depleted in $P_1f_3$.

**Table 1.** Some abundant major element oxides of these samples.

| Formation | $SiO_2$ | Average | $Al_2O_3$ | Average | CaO | Average |
|-----------|---------|---------|-----------|---------|-----|---------|
| $P_1f_1$ | 46.88–70% | 57.47% | 3.85–8.49% | 5.6% | 4.59–8.02% | 6.4% |
| $P_1f_2$ | 41.67–75.73% | 54.83% | 2.82–11.22% | 7.96% | 2.1–13.07% | 7.43% |
| $P_1f_3$ | 52.6–79.15% | 60.87% | 2.98–15.59% | 10.52% | 1.43–10.43% | 5.24% |

The Al/(Al + Fe + Mn) ratio can be used as an indicator for hydrothermal activity. For $P_1f$, the ratio ranged from 0.6 to 0.81, with an average of 0.7. Based on the abnormal peak in the Al/(Al + Fe + Mn) ratio of the Fengcheng formation, two phases of hydrothermal

activity occurred during the early and middle depositional period of $P_1f_2$. This result is consistent with previous studies [29].

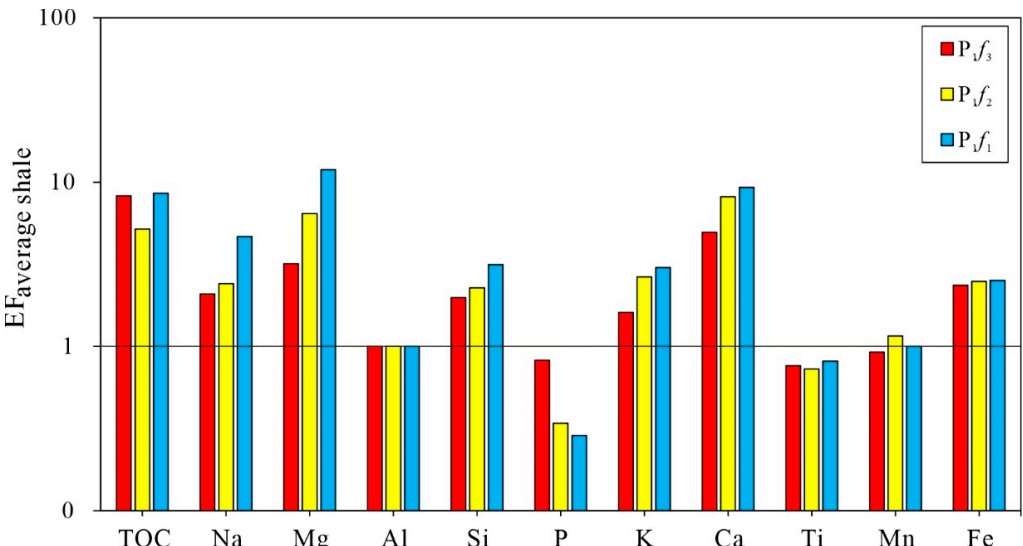

**Figure 3.** The plot for concentration factors ($EF_{average\ shale}$) of TOC and major elements of the Lower Permian Fengcheng formation in Mahu Sag, Junggar Basin. The horizontal solid line represents $EF_{average\ shale}$ = 1 to assess enrichment or depletion of elements. Average shale data are from Wedepohl [30].

The average concentrations of trace elements Sr, Cu, Rb, Th, U, V, Ni, Ba, Cr, Zr, Mo, and Sc were 392.53, 36.09, 51.27, 2.5, 3.07, 102.48, 37.59, 246.04, 45.2, 107.94, 20.47, and 8.25, respectively, as shown in Figure 4. The most abundant trace elements were V, Sr, Zr, and Ba, all were more than 100 ppm, while the elements Th, Sc, and U were less than 10 ppm. Relative to average shale, the samples were obviously enriched in Sr and Mo, and slightly enriched in V, Ni, U, Cu, Ba, Zr, and Sc, and depleted in Cr and Th.

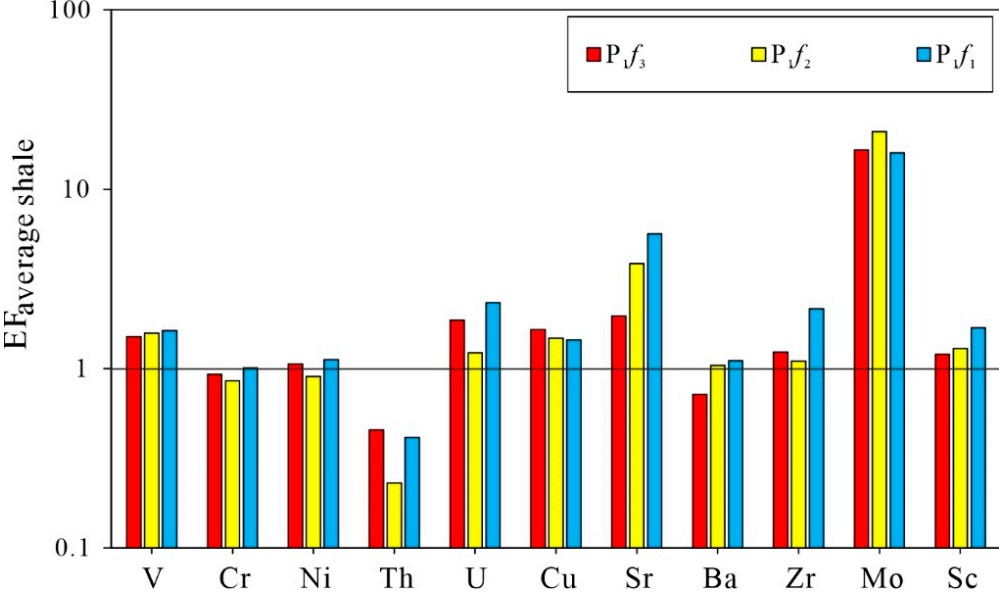

**Figure 4.** The plot for concentration factors ($EF_{average\ shale}$) of trace elements of the Lower Permian Fengcheng formation in Mahu Sag, Junggar Basin. The horizontal solid line represents $EF_{average\ shale}$ = 1 to assess the enrichment or depletion of elements. Average shale data are from Wedepohl [30].

### 4.2. Contents of Rare Earth Elements

The samples from $P_1f_2$ and $P_1f_3$ had similar PAAS normalized patterns, while the samples from $P_1f_1$ had different patterns. The REE contents of the samples from the Fengcheng formation ranged from 33.7 ppm to 207.34 ppm, with an average of 90.03 ppm, which is lower than the PAAS value, with a lower value of REE contents in all samples compared to PAAS. Compared with HREE, LREE were relatively enriched, as shown in Table 2. The calculated results of $(La/Yb)_N$ also show a remarkable concentration in LREE over HREE. In the study area, $(La/Sm)_N$ exhibits a wider distribution in comparison of $(Gd/Yb)_N$, which implies that the fractionations in LREE are larger than that in HREE. The PAAS normalized REE patterns of samples from the Fengcheng formation are all characterized by LREE and comparatively flat HREE trends, as shown in Figure 5. The calculated results display low Ce anomalies; the Ce/Ce* ratios are shown in Table 2.

**Table 2.** Some rare earth element oxides of these samples.

| Formation | $\frac{\sum LREE}{\sum HREE}$ | Average | $(La/Yb)_N$ | Average | $(La/Sm)_N$ | Average | $(Gd/Yb)_N$ | Average | Ce/Ce* | Average |
|---|---|---|---|---|---|---|---|---|---|---|
| $P_1f_1$ | 3.17–4.77 | 3.74 | 0.59–1.04 | 0.77 | 0.62–0.84 | 0.74 | 0.8–1.27 | 0.95 | 0.98–0.99 | 0.99 |
| $P_1f_2$ | 0.47–6.63 | 2.77 | 0.07–1.3 | 0.71 | 0.41–1.37 | 0.75 | 0.28–3.1 | 1.17 | 0.84–1.04 | 0.97 |
| $P_1f_3$ | 1.51–6.46 | 2.38 | 0.31–1.56 | 0.57 | 0.41–0.89 | 0.58 | 0.59–1.88 | 1.05 | 0.8–1.04 | 0.96 |

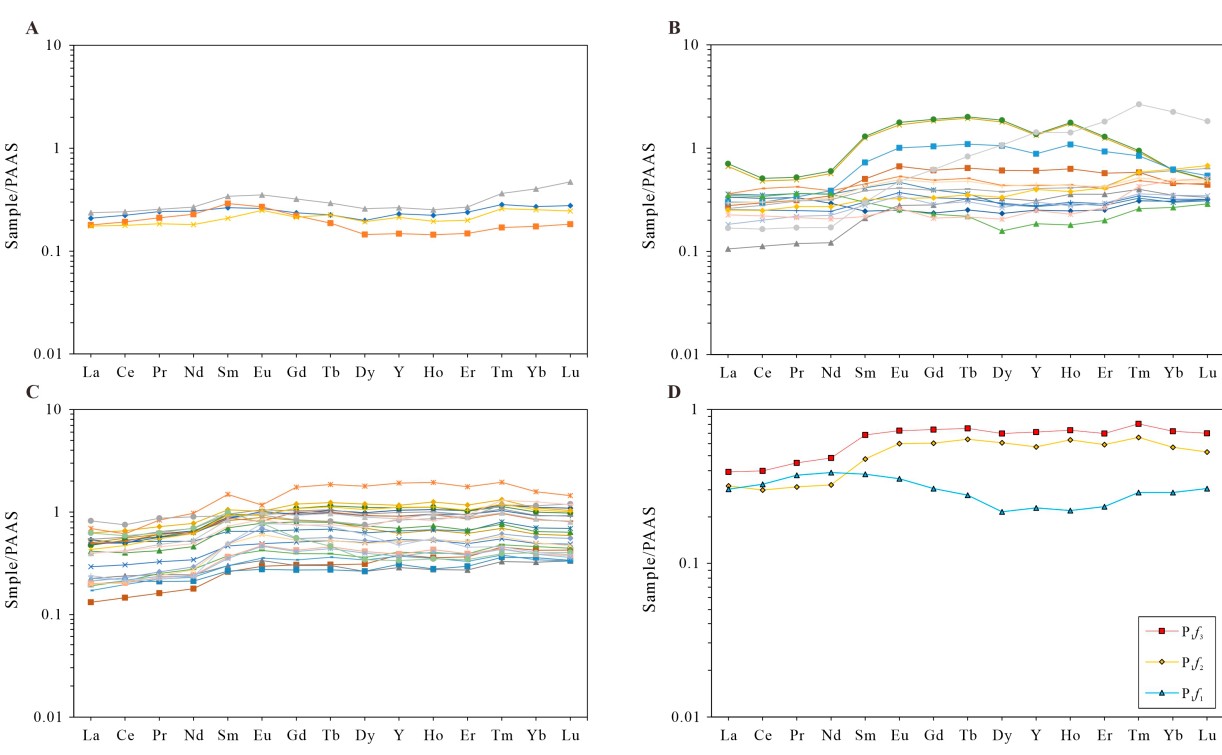

**Figure 5.** Plot of Post Archean Australian Shale-normalized patterns of (**A**) samples from $P_1f_1$; (**B**) samples from $P_1f_2$; (**C**) samples from $P_1f_3$; (**D**) average values of $P_1f_1$, $P_1f_2$, and $P_1f_3$ in Mahu Sag, Junggar Basin.

## 5. Discussion

### 5.1. Contents of Total Organic Matter

In organic geochemical studies, the values of total organic carbon (TOC) are usually used to assess the organic matter abundance of source rock. The value of the TOC parameter is the direct result of enrichment or depletion of organic matter. In Mahu Sag, the average TOC contents of the samples from $P_1f_1$, $P_1f_2$, and $P_1f_3$ are 0.15–0.73% (with an average of 0.53%), 0.03–1.73% (with an average of 0.42%), and 0.01–3.43% (with an average of 0.67%),

respectively. Figure 3 shows that the TOC content of samples from Fengcheng formation is enriched relative to common shale. For $P_1f_1$, $P_1f_2$ and $P_1f_3$, the average EF values are 8.56, 5.18, and 8.24, respectively.

### 5.2. Analysis of Depositional Environments

#### 5.2.1. Paleoclimate

As mentioned in [26], the change in paleoclimate resulted in enrichment or depletion of trace elements. The C-value is a good indicator of paleoclimate, and can be calculated using the following formula: C-value = (Fe + Mn + Cr + Ni + V + Co)/(Ca + Mg + Sr + Ba + K + Na) [31–33]. Because Fe, Mn, Cr, Ni, V, and Co are commonly concentrated under humid conditions, and Ca, Mg, Sr, Ba, K, and Na are enriched under arid conditions, the C-value may increase due to the climate changing from arid to humid. According to results, the average C-values of $P_1f_1$ and $P_1f_2$ are 0.18 (0.14–0.23) and 0.23 (0.13–0.49), indicating a semiarid paleoclimate, and the paleoclimate of $P_1f_2$ was more arid compared to that of $P_1f_1$. The C-value of $P_1f_3$ is 0.2–1.13 (with an average of 0.44), reflecting a semiarid to semi-humid environment during the depositional period.

As described in [11], the ratios of Sr/Cu are very sensitive to changes in the paleoclimate. The Sr/Cu ratios ranging from 1.3 to 5.0 indicate that the climate was humid and warm; the ratios ranging from 5.0 to 10 indicate a semiarid and semi-humid climate; the ratios greater than 10 indicate a hot and arid climate [34,35]. The ratios of Sr/Cu in $P_1f_1$ ranged from 10.91 to 43.25, all greater than 5.0, with an average value of 28.94. The average Sr/Cu ratio of $P_1f_2$ was 18.21 (ranging from 3.89 to 43.08), and the Sr/Cu ratios in $P_1f_2$ were mostly greater than 5.0 (except for sample F7-34). The ratios of Sr/Cu in $P_1f_3$ ranged from 2.71 to 19.71, with an average of 8.52, and two-thirds of the samples from $P_1f_3$ had Sr/Cu > 5.0. The average Sr/Cu values of the samples dropped from $P_1f_1$ to $P_1f_3$, reflecting that the paleoclimate gradually became relatively humid from the early to the late depositional period. In the late depositional period of $P_1f_3$, the values of Sr/Cu changed frequently, indicating that the sedimentary paleoclimate of the Fengcheng formation alternated frequently from arid to humid, as shown in Figures 6 and 7.

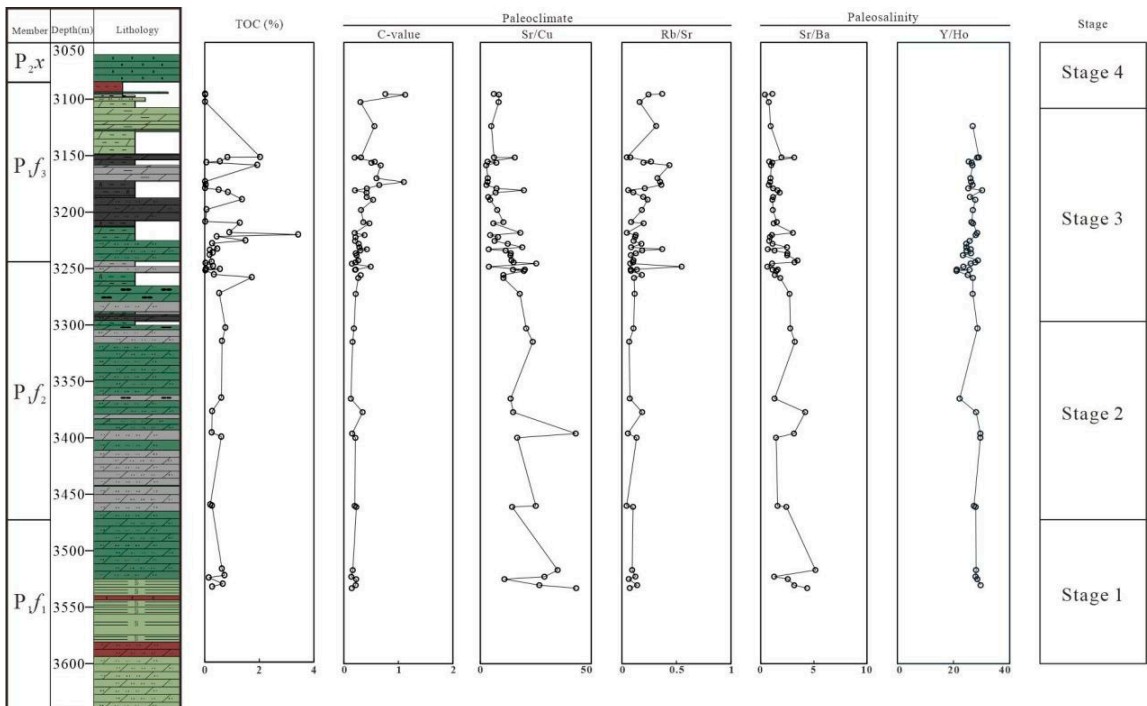

**Figure 6.** Vertical change in TOC, paleoclimate indicators (C-value, Sr/Cu, and Rb/Su) and paleosalinity indicators (Sr/Ba and Y/Ho).

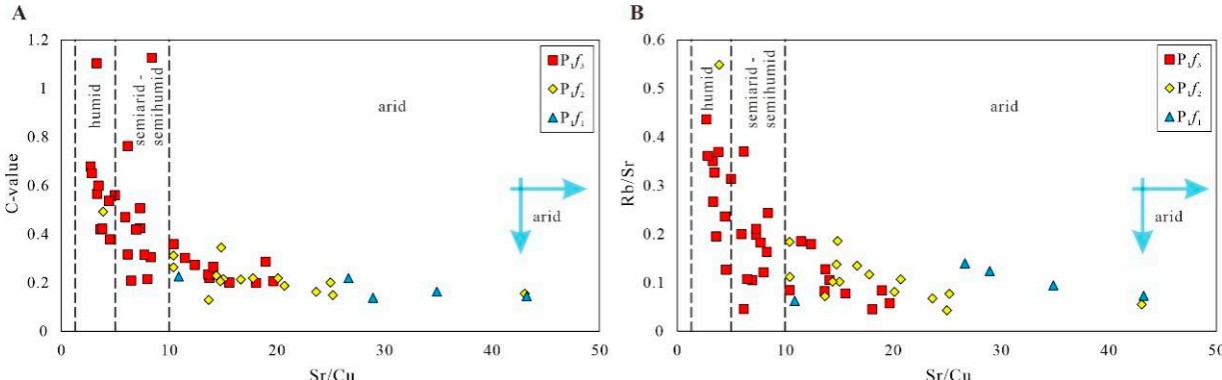

**Figure 7.** The cross plots of element indicators for paleoclimate. (**A**) Plot of Sr/Cu vs. C-value; (**B**) Plot of Sr/Cu vs. Rb/Sr.

The ratio of Rb/Sr is also an important indicator of paleoclimate because Rb is relatively stable during weathering, while Sr is lost during leaching. When the climate was humid, precipitation increased, and weathering intensity became stronger, resulting in a loss of Sr and an increase in the Rb/Sr value. When the climate was dry, the precipitation decreased, the weathering decreased, and the rocks were relatively rich in Sr; the Rb/Sr ratio would be relatively low [36,37]. A low Rb/Sr value means a dry climate and a high value means a humid climate. The change ranges of $P_1f_1$, $P_1f_2$, and $P_1f_3$ were 0.06–0.14 (average 0.10), 0.04–0.55 (average 0.13), and 0.05–0.44 (average 0.19), respectively. This indicates that the climate of the depositional period of the Fengcheng formation was predominantly dry, and changed frequently in the late depositional period of $P_1f_2$ and $P_1f_3$, as shown in Figures 6 and 7.

### 5.2.2. Paleo-Salinity

Paleo-salinity may affect the stratification of water bodies, which in turn affects the degree of paleo-productivity and preservation of organic matter [38]. The Sr/Ba ratio is regarded as a paleo-salinity indicator [39–42]. The chemical properties of Sr are similar to those of Ba, and the solubility of Sr is greater than that of Ba [43]. Ba precipitates as $BaSO_4$ at lower salinity levels than Sr to form $SrSO_4$. Therefore, with increasing salinity of water, Ba precipitates first in the form of $BaSO_4$, and Sr is enriched in water. As salinity increases continuously, Sr precipitates to form $SrSO_4$. Therefore, a higher Sr/Ba value indicates higher salinity. When the Sr/Ba ratio is less than 0.6, it indicates a freshwater environment; when the Sr/Ba ratio is greater than 1.0, it indicates a saltwater environment; when the Sr/Ba ratio ranges from 0.6 to 1.0, it indicates a brackish water environment. As shown in Table 3, the largest average Sr/Ba ratio occurred in $P_1f_1$, varying from 1.26 to 5.15 (with an average of 3.3). The Sr/Ba ratios of $P_1f_2$ ranged from 0.63 to 4.19, with an average of 2.07. The Sr/Ba ratios of $P_1f_3$ ranged from 0.4 to 3.47 (with an average of 1.43), and the Sr/Ba ratios of two thirds of the samples were more than 1, indicating that the water bodies were in brackish to saltwater environments. The change in the Sr/Ba ratio indicates that the water body of Fengcheng formation was mainly in a saltwater environment during the depositional time, and the overall paleo-salinity of the water body was high. The Y/Ho ratio is generally a good indicator, and a high value indicates a seawater environment [44]. The average ratios of Y/Ho were 28.53 (ranging from 27.9 to 29.67), 26.1 (ranging from 21.09 to 29.6), and 26.63 (ranging from 23.37 to 30.25) for $P_1f_1$, $P_1f_2$, and $P_1f_3$, respectively. Apart from a few abnormal values, the paleo-salinity of water bodies decreased gradually during the depositional period of the Fengcheng formation, as shown in Figures 6 and 8.

**Table 3.** Sr/Ba ratio in samples from the Fengcheng formation in Mahu Sag.

| Formation | Y/Ho | Average | Sr/Ba | Average | Lithology |
|---|---|---|---|---|---|
| $P_1f_3$ | 23.37–30.25 | 26.63 | 0.4–3.47 | 1.43 | Brackish water environment |
| $P_1f_2$ | 21.09–29.6 | 26.1 | 0.63–4.19 | 2.07 | Brackish water environment |
| $P_1f_1$ | 27.9–29.67 | 28.53 | 1.26–5.15 | 3.3 | Brackish water environment |

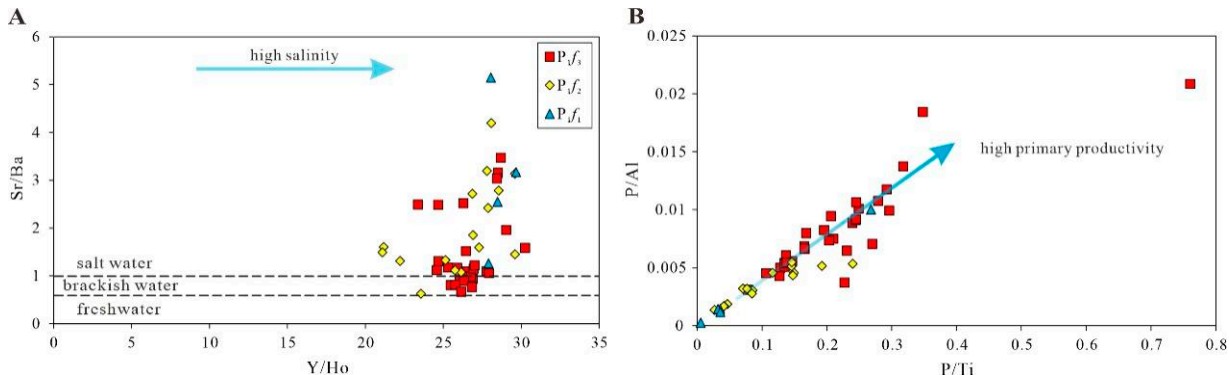

**Figure 8.** The cross plots for reflection of paleo-salinity, Y/Ho vs. Sr/Ba (**A**) and paleo-productivity, P/Ti vs. P/Al (**B**).

The change in evaporation and rainfall might lead directly to the change in the salinity of water bodies. Therefore, the paleo-salinity can reflect the change in the paleoclimate to a certain degree [45]. As mentioned in [11,33], in warm–humid climate conditions, due to the injection of massive amounts of freshwater, the salinity of water bodies decreases, while under a hot–arid climate, due to the evaporation of large amounts of fresh water, the salinity of water bodies increases. In the study area, the change trend of the salinity of paleo-sedimentary water bodies is basically consistent with that of the paleoclimate, which also reflects that the results of the paleoclimate analysis of the Fengcheng formation are correct. Figure 9 shows that the high Sr/Ba ratios are mainly located in the high-Sr/Cu-ratio zone and low-C-value zone, indicating that the paleoclimate was the main factor affecting the paleo-salinity of water bodies.

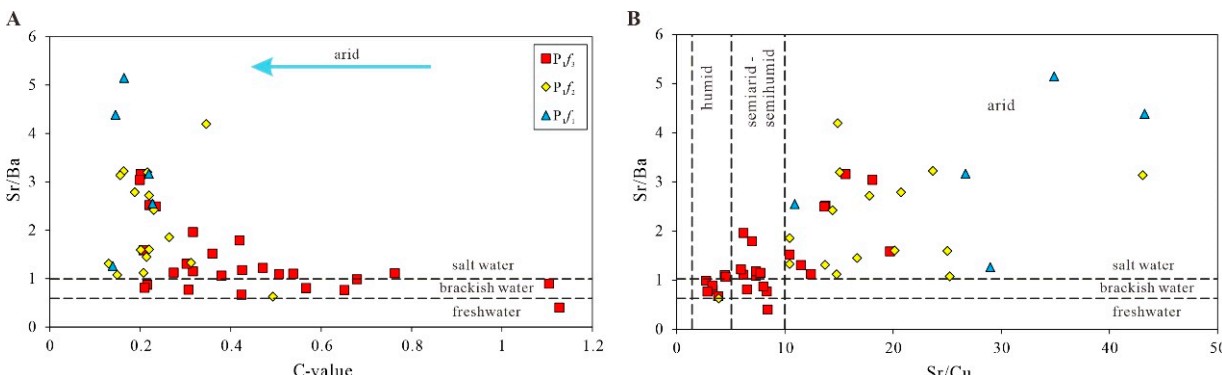

**Figure 9.** The cross plots of C-value vs. Sr/Ba (**A**) and Sr/Cu vs. Sr/Ba (**B**) of the Fengcheng formation in Mahu Sag.

### 5.2.3. Paleo-Redox Conditions

The ratio of redox-sensitive elements is one of the most widely used indicators of redox conditions in sedimentary systems [46]. Redox-sensitive elements are easily soluble under oxidizing conditions, and less soluble under reducing conditions, being enriched in

an oxygen-depleted sedimentary environment. Furthermore, they are generally immobile during the diagenesis process, keeping the original records during deposition. Therefore, redox-sensitive elements have been used in recording the change in redox environments of water bodies, and quantitative identification standards have been widely used [47]. The ratios of redox-sensitive elements, such as V/Cr, V/(V + Ni) and U/Th, are relatively reliable parameters, and are usually used as elemental indicators for distinguishing the anoxic, dysoxic, and oxic conditions of depositional environments [48]. A U/Th ratio of less than 0.75 indicates oxic conditions; a ratio greater than 1.25 indicates anoxic conditions; and a value ranging from 0.75 to 1.25 indicates dysoxic conditions [49]. When the V/Cr ratio is greater than 4.25, it indicates an anoxic environment; when the V/Cr ratio is less than 2, it indicates an oxic environment; when the ratios range from 2 to 4.25, an dysoxic environment is indicated. The analyzing data indicate that, when the V/(V+Ni) ratio is less than 0.6, the sedimentary conditions are oxic; when the ratio is greater than 0.77, the sedimentary conditions are anoxic; when the ratio ranges from 0.6 to 0.77, the sedimentary environments are dysoxic. The data from the samples of $P_1f_1$ and $P_1f_2$ are shown is Table 4. As shown in Figures 10 and 11, these results indicate that the redox conditions of the water body gradually changed from anoxic–anoxic to anoxic–anoxic during the entire deposition stage, and the redox conditions change frequently in $P_1f_3$.

**Table 4.** Redox-sensitive elements in samples from Fengcheng formation in Mahu Sag.

| Formation | V/Cr | Average | V/(V + Ni) | Average | U/Th | Average |
|---|---|---|---|---|---|---|
| $P_1f_1$, $P_1f_2$ | 1.75–4.93 | 2.56 | 0.69–0.85 | 0.76 | 0.74–3.37 | 1.81 |
| $P_1f_3$ | 0.95–5.74 | 2.33 | 0.4–0.94 | 0.71 | 0.21–3.08 | 1.33 |

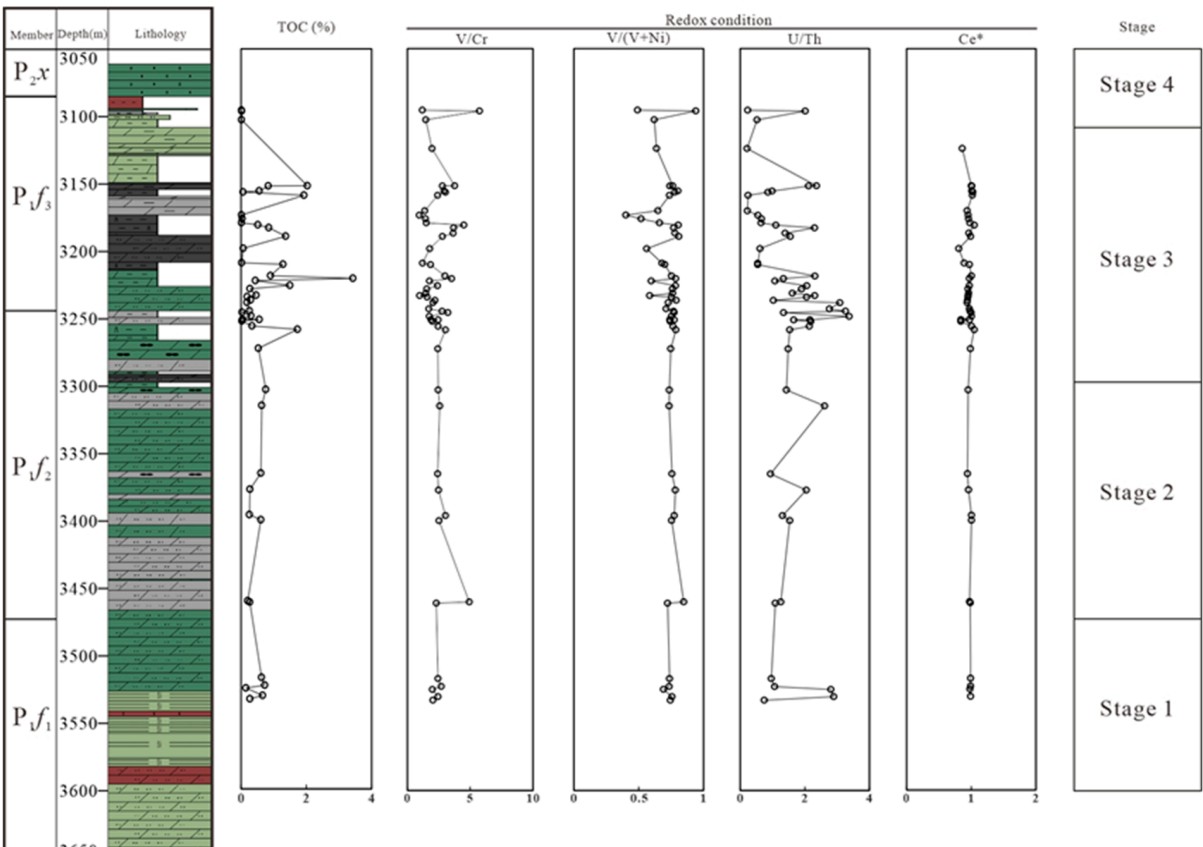

**Figure 10.** Vertical change of TOC and redox condition parameters (V/Cr, V/(V + Ni), U/Th, and Ce*).

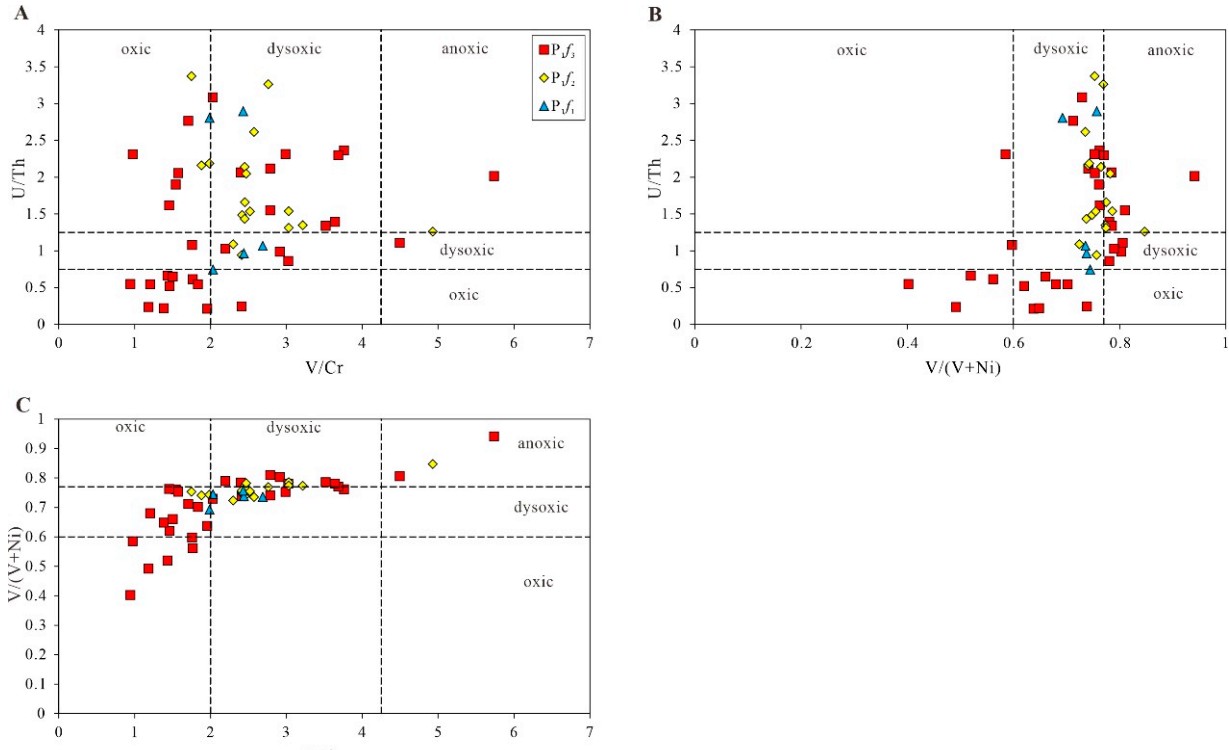

**Figure 11.** The cross plots of U/Th vs. V/Cr (**A**), U/Th vs. V/(V + Ni) (**B**), and V/(V + Ni) vs. V/Cr (**C**) of the samples from the Fengcheng formation in Mahu Sag.

In addition, $Mo_{EF}$ and $U_{EF}$ are relatively new and particularly useful for redox analysis in sedimentary systems [50]. In the studied area, almost all samples display a similar pattern with Mo content clearly higher than U content (except for four samples from $P_1f_3$). Figure 12 shows that most samples are located in the zone around the green field, and the concentration pattern deviates from the gray field (unrestricted marine). This enrichment pattern is consistent with the presence of intense redox cycling of the metal-oxyhydroxide particulate shuttle process within the water body ranging from anoxic to strong euxinic. In contrast, as shown in Figure 12, only four samples fall into the bottom of the gray field, suggesting transient suboxic conditions for the water body during the depositional period of $P_1f_3$. According to the Mo-U pattern, the water body changed gradually from weakly restricted to unrestricted conditions during the late depositional period of $P_1f_3$, indicating the lake level rising due to seasonal paleoclimate.

Ce anomaly can also be used as a geochemical indicator for describing the redox conditions in the depositional environment due to its sensitivity to redox conditions [51]. Under oxic conditions, the solubility of $Ce^{4+}$ is very low in the sea, resulting in the relative depletion of Ce in the water body, and a positive Ce anomaly. As shown in Figure 10, the calculated results of all samples display weak anomalies, and the redox conditions are not consistent with conditions obtained from other parameters. This result may be affected by potential contaminations from widespread leaching of REE from other minerals and organic matter.

### 5.2.4. Primary Productivity

Phosphorus can be used as a trace element indicator for estimating levels of primary productivity. P is considered to be an essential nutrient element in primary production. By eliminating the influence of terrigenous detrital materials, P/Ti and P/Al are good indicators of paleo-productivity. High ratios indicate high paleo-productivity [52]. The $P_1f_3$ had the highest P/Ti ratio in all parts of the Fengcheng formation, ranging from 0.08 to 0.76 (with an average of 0.22). The $P_1f_2$ had a ratio of 0.03–0.24 (with an average of 0.1),

while the primary productivity of $P_1f_1$ was lower than that of $P_1f_2$, and the ratio of $P_1f_1$ ranged from 0.01 to 0.27 (with an average of 0.08). According to the calculated results, the paleo-productivity decreased gradually during the depositional period of the Fengcheng formation, indicating that the paleo-productivity of the water body at the shore of lake was relatively low during the deposition of the Fengcheng formation. As shown in Figure 13, the vertical trend of the P/Al ratio is similar to that of P/Ti. The P/Al of the Fengcheng formation are in the order of $P_1f_3$ (0.0083) > $P_1f_2$ (0.0034) > $P_1f_1$ (0.0029), with the ratios ranging from 0.0031 to 0.0209, 0.0012 to 0.0055, and 0.0003 to 0.01, respectively. As shown in Figure 8B, the ratio characteristics also show a relatively poor primary productivity in the Fengcheng formation.

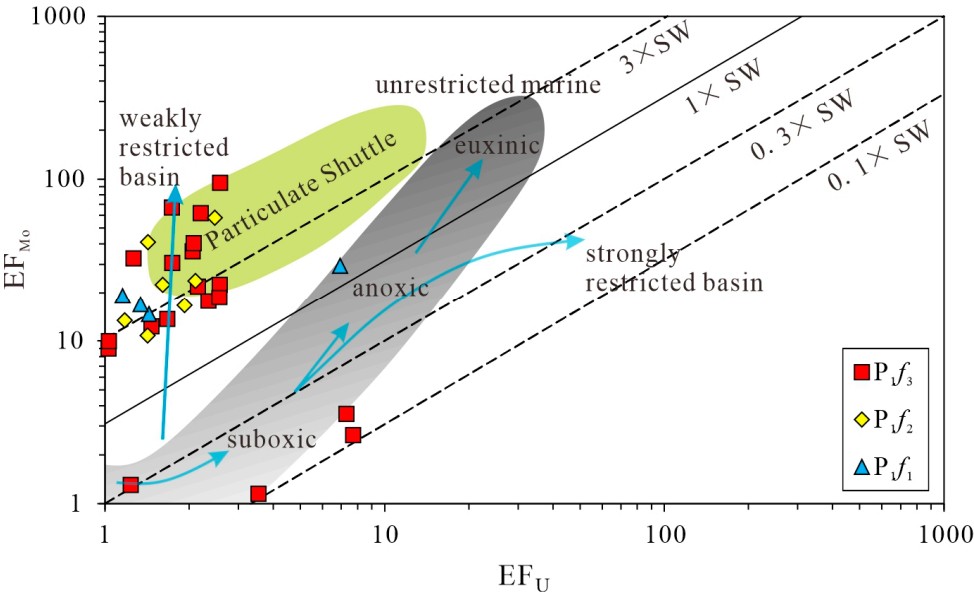

**Figure 12.** The cross plot of $EF_{Mo}$ versus $EF_U$ for the samples from the Fengcheng formation normalized by average shale. The lines show Mo/U molar ratios relative to present-day seawater value. Dotted lines represent the Mo/U molar ratios equal to multiples of present-day seawater value (0.1 × SW, 0.3 × SW, 3 × SW), and the solid line indicates a Mo/U molar ratio equal to seawater value (SW). The gray field represents the "unrestricted marine" trend, whereas the green field represents the "particulate shuttle" trend.

5.2.5. Detrital Influx

Detrital input can affect the depositional rate and biodegradation efficiency of organic matter. Detrital material is transported from the continents into the oceans primarily by continental runoff and fluvial systems, and atmospheric dust also contributes [53]. Titanium (Ti) and Aluminum (Al) are the main component of the continental crust. Therefore, the contents of Ti and Al in marine sediments are often used to analyze the contribution of detrital input. The content of Ti is low relative to average shale in all samples, with an EF lower than 1. The relatively good correlation between $TiO_2$ and $Al_2O_3$ ($r^2$ = 0.7154, shown in Figure 14A) suggests that either Ti is found within clay lattices, or the detrital material came from a constant source. Ti and Al are mainly from detrital provenance, and immobile during diagenesis. Therefore, the Ti/Al ratio is commonly considered to be an indicator for determining terrigenous detrital input. High Ti/Al ratios reflect detrital input and large grains.

According to the analysis, the ratios of Ti/Al had similar distributions at different intervals. In addition, the contents of Th, Zr, and Sc are also used to estimate detrital input, due to higher amounts of these in average shale than in marine carbonate [54]. Figure 14 shows a weak-to-moderate positive correlation between Th and ∑REE ($r^2$ = 0.5298), and a weak positive correlation for Zr vs. ∑REE ($r^2$ = 0.0983) and Sc vs. ∑REE ($r^2$ = 0.328). Although the correlation between Th and ∑REE is slightly moderate, the weak correlation

between Zr and Sc and ∑REE reflects limited detrital input during the depositional period of the Fengcheng formation.

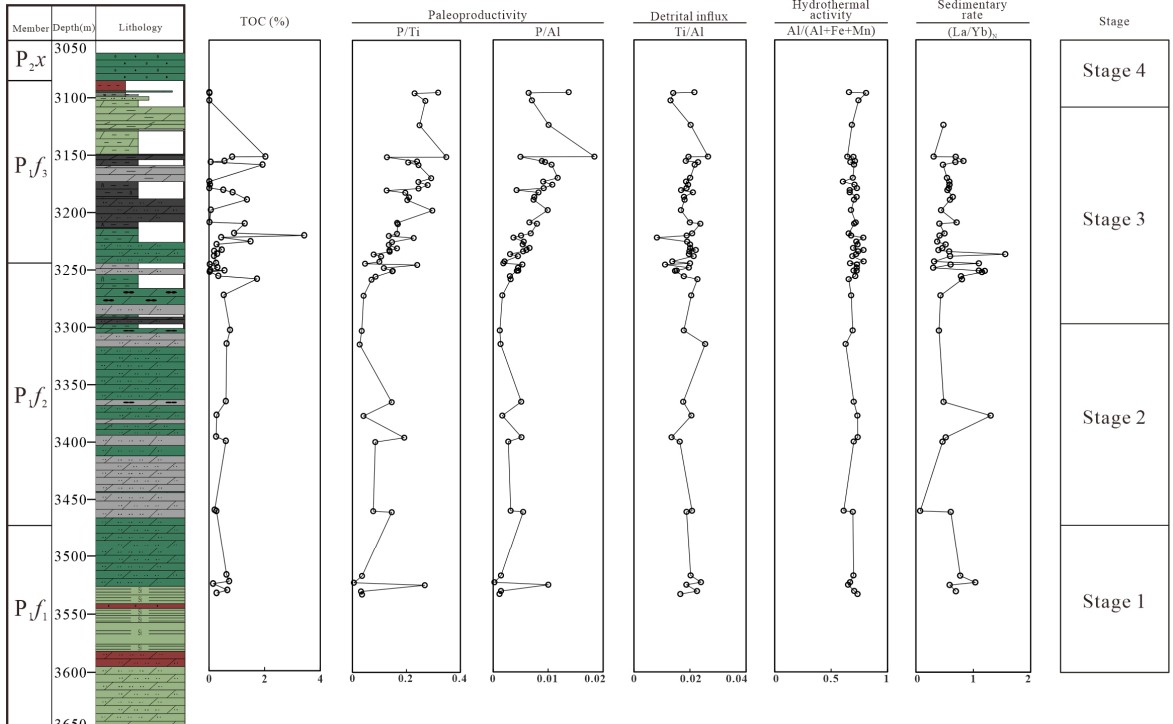

**Figure 13.** The vertical change in TOC, paleo-productivity indicators (P/Ti and P/Al), detrital influx indicator (Ti/Al), hydrothermal activity indicator (Al/(Al + Fe + Mn)), and sedimentary rate indicator ((La/Yb)$_N$).

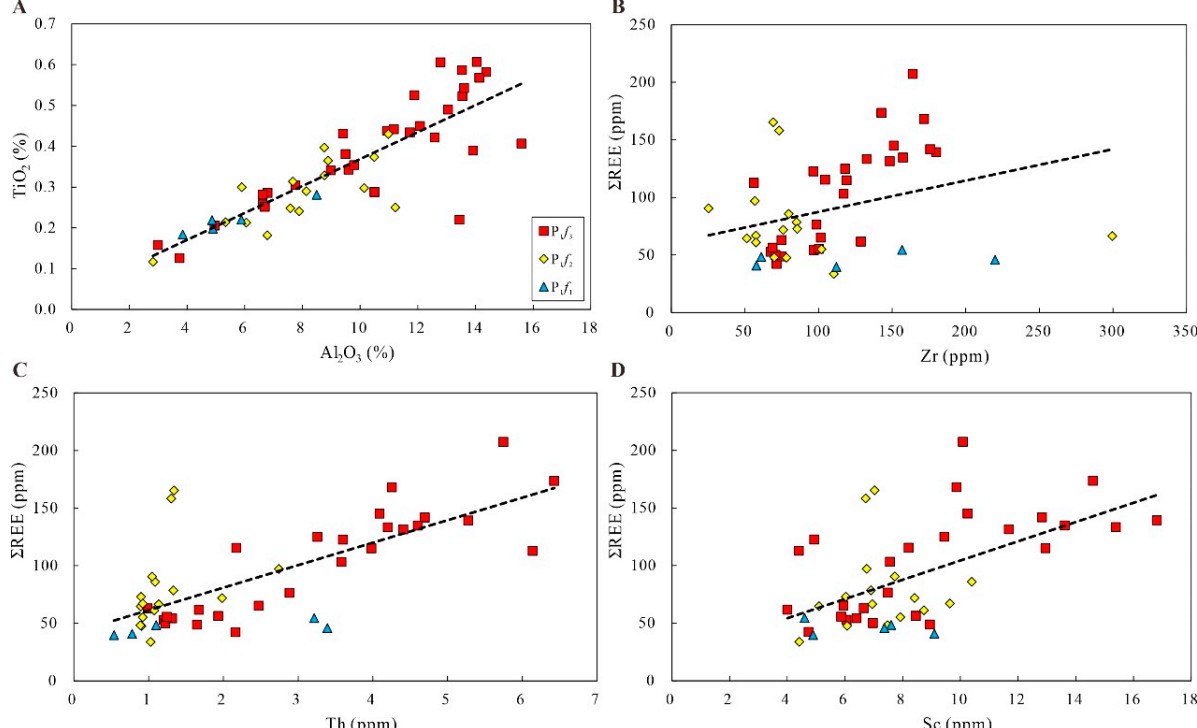

**Figure 14.** Plot of Al$_2$O$_3$ content versus TiO$_2$ content (**A**), and correlations between ∑REE and Zirconium (**B**), Thorium (**C**), and Scandium (**D**) used to estimate input.

*5.3. Control Factors for Organic Matter*

Primary productivity provides the material source for organic matter [55]. In the $P_1f_1$ and the early–middle interval of $P_1f_2$, the volcanic ash released by the frequent volcanic activities in the surroundings accelerated the growth of lake surface producers, supplying them with ample material sources [56]. After the middle stage of $P_1f_2$, the paleo-productivity was relatively high, corresponding to high redox condition indicators. This result indicates that a large amount of oxygen was consumed in the water body with the increase in primary productivity, resulting in a dysoxic environment in the depositional period. On the other hand, due to dysoxic conditions and high primary productivity, the abundant organic matter has been preserved effectively, resulting in a relatively high TOC value. The high paleo-productivity and dysoxic conditions are two favorable factors for the accumulation of organic matter. Additionally, P/Al ratios and P/Ti ratios increased, with similar vertical trends to TOC, as shown in Figure 13. However, as shown in Figure 15, TOC is weakly negatively correlated with paleo-productivity, indicating that paleo-productivity is not the only controlling factor for enrichment of organic matter.

The preservation model of organic matter can be established by the two main factors and several secondary factors. Anoxic conditions are the vital preservation conditions for the accumulation of organic matter. The water body under anoxic conditions was favorable for preferential preservation of organic carbon. Therefore, the concentration of organic matter is a result of enhanced preservation potential under anoxic conditions. Figure 15 shows the cross-plots of TOC and redox indices (V/Cr, V/(V + Ni), U/Th), which were drawn to estimate the correlation between accumulation of organic matter and redox conditions. As shown in Figure 10, although the redox indices have similar increasing vertical trends to TOC, the detailed trends are not well coupled. TOC is weakly positively correlated with V/Cr, V/(V + Ni), and U/Th, indicating that redox conditions were not the main controlling factor for the accumulation of organic matter.

Terrigenous detrital influx is another main factor for the preservation model. Detrital input is favorable for the burying and preservation of organic matter. However, terrigenous detrital input also has a diluting effect on the enrichment of organic matter. Ti/Al, an indicator for terrigenous influx, indicates that the detrital input in the Mahu Sag was limited but fluctuated during the depositional period of the Fengcheng formation. In the late $P_1f_1$, the TOC was obviously positive correlated with Ti/Al, reflecting that the terrigenous materials have a positive role in the enrichment of organic matter. Afterwards, the lake level declined, and the input of detrital materials decreased gradually. In general, during the depositional period of $P_1f$, the terrigenous input was always low, indicating a rising lake level and a stable sedimentary environment. However, as shown in Figure 15, TOC is weakly positively correlated with the parameters, indicating that terrigenous influx had no obvious dilution/preservation effect on the accumulation of organic matter. That is to say, detrital input is not the primary factor affecting the accumulation of organic matter.

High salinity is not favorable for the proliferation of microorganisms because the number of species which are able to adapt to high-salinity water is limited [57]. Under the paleoclimate conditions of $P_1f_3$, due to the injection of a great quantity of freshwater, the salinity decreased, accelerating the blooms of the microorganisms in the surface water. Therefore, compared with the Lower Permian $P_1f_3$ with lower salinity, the primary productivity of microorganisms was lower in $P_1f_1$ and $P_1f_2$, which are high in salinity. Despite the TOC content of $P_1f$ increasing under a decreasing paleo-salinity environment with a more humid paleoclimate, the correlation between TOC and factors (paleoclimate, paleo-salinity) is not obvious, illustrating that the concentration of organic matter is also controlled by other factors, as shown in Figure 15.

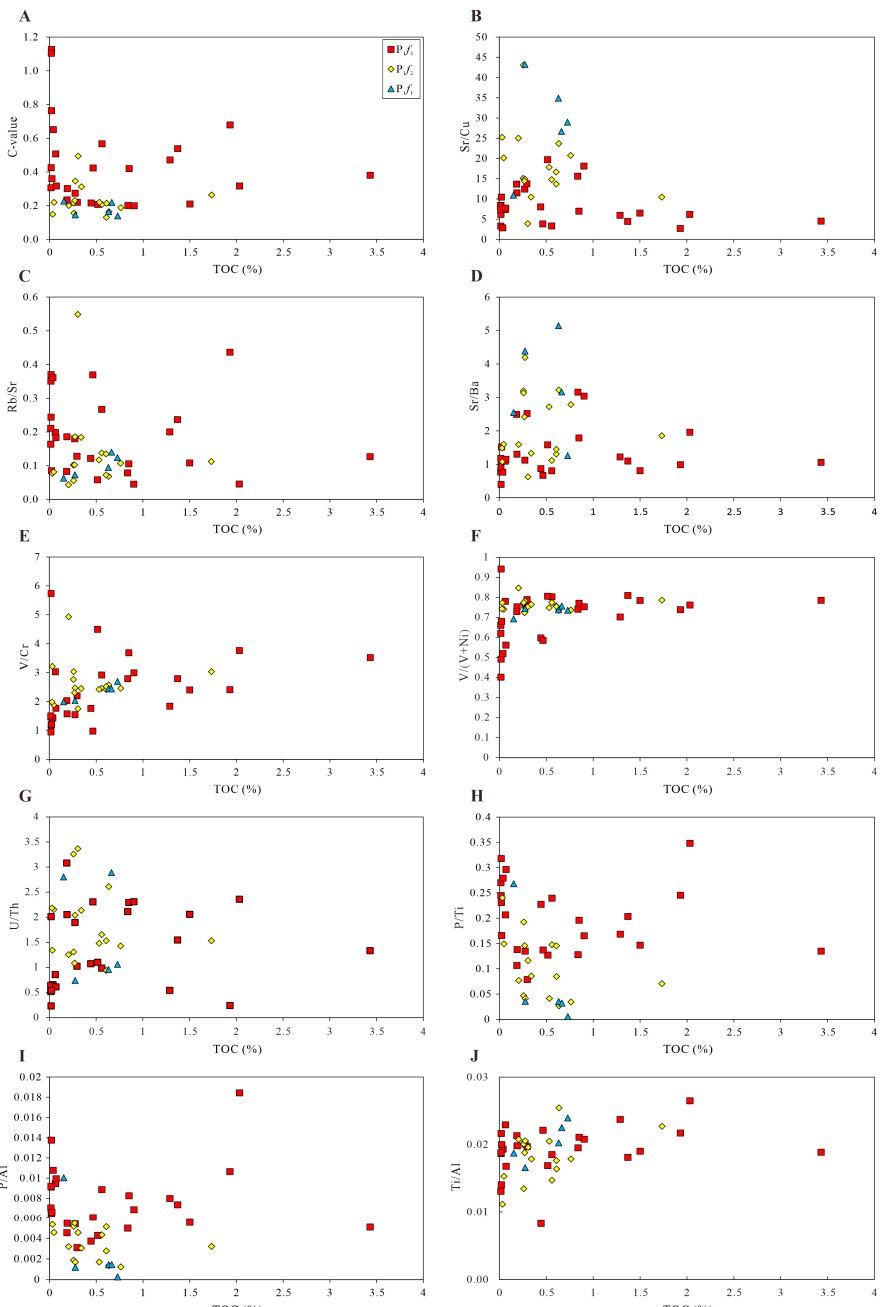

**Figure 15.** Correlations between TOC and several parameters, such as (**A–C**) paleoclimate indicators (C-value, Sr/Cu, and Rb/Sr), (**D**) paleo-salinity indicator (Sr/Ba), (**E–G**) redox condition parameters (V/Cr, V/(V + Ni), and U/Th), (**H,I**) paleo-productivity indicators (P/Ti and P/Al), and (**J**) detrital influx indicator (Ti/Al).

Generally, the concentration of organic matter was controlled by the paleoclimate, paleo-salinity, redox conditions, paleo-productivity, and detrital input. There was no single factor that dominantly controlled the concentration of organic matter. In other words, the combination of preservation and production conditions controlled the accumulation of organic matter in the Lower Permian $P_1f$, rather than it being controlled only by preservation conditions or production conditions. The effect of redox conditions is weaker than that of others factors. Based on the absolute value, from low to high, the partial correlation coefficient of the controlling factors are, successively, redox conditions, paleoclimate, terrigenous input, paleo-salinity, and paleo-productivity.

*5.4. Formation Model*

Based on the above study, the depositional period of the Fengcheng formation can be divided into four stages. The complete sedimentary evolution of alkaline lakes can be divided into four stages, including the early onset stage, preliminary stage, strong stage, and weak terminal stage of alkalinity. Except for the strong stage of alkalinity, all the stages can be observed in $P_1f$ in Well F7. Stage 1 and stage 2 correspond to the early onset stage and preliminary stage, respectively, while stages 3 and 4 are in the weak terminal stage of alkalinity.

Stage 1 is at the late depositional stage of $P_1f_1$, characterized by dysoxic conditions, arid environment with high salinity, and relatively low paleo-productivity. The appearance of andesite indicates that the deposition of $P_1f_1$ was influenced by surrounding volcanic activities at the early stage. At the middle of stage 1, there seems to have been a short period of rainfall, and the lake level was rising. The detrital input was positively correlated with TOC, suggesting that the terrigenous materials provided nutrients for the accumulation of organic matter. At the same time, the water body became more reductive and the paleo-productivity increased. However, the paleoclimate transformed from arid to humid, with decreasing salinity and rapid depositional rate, which was not favorable for the preservation of organic matter, resulting in a low content of organic matter at this stage.

Stage 2 corresponds to the early–middle depositional stage of $P_1f_2$. Compared with stage 1, under the conditions of an anoxic climate, the redox condition of this stage is more obvious. Under arid conditions, water evaporation is greater than precipitation, resulting in the falling of the lake level and an increase in salinity. Meanwhile, at the early–middle stage, the primary productivity was moderate, and the influence of terrigenous influx on the water body was low. Earthquakes are considered to be one of the triggering mechanisms for the deformation of structures of soft sediments at the bottom of water bodies [58]. Previous studies have suggested that seismic activity was frequent during the depositional period of $P_1f_2$ based on the continuous vertical and lateral distribution characteristics of the deformation structure of soft sediment. Deep hydrothermal fluids intruded upward along the fault and were affected by frequent seismic activity. According to the above discussion of geochemical characteristics, there were two phases of hydrothermal activity, which occurred at the early and middle–late sedimentary stage of $P_1f_2$, respectively. Compared with the hydrothermal activity at the first stage, the organic matter accumulated in the second stage of the hydrothermal event was more abundant. The strong hydrothermal activity triggered the concentration of abundant reducing gases and minerals. However, the relatively hot and dry climate resulted in a low input of nutrients, preventing algal blooms on a large scale. Furthermore, during the two stages of hydrothermal activity, high concentrations of $SiO_2$ diluted the primary productivity, resulting in the low TOC values of the hydrothermal sediments.

Stage 3 is from the late $P_1f_2$ to middle stage of $P_1f_3$. In this depositional period, the paleoclimate showed seasonally frequent fluctuation from semiarid to semi-humid, and generally was more warm and humid than that of stage 2. In the lacustrine basin, the change in the lake level was closely related to climate change at smaller scales. The paleo-salinity of the water body also changed drastically under the seasonally repeated climate change, showing a gradually decreasing trend on the whole. The terrigenous input was negatively correlated with paleoclimate, indicating that paleoclimate is not the main controlling factor for terrigenous input. During this depositional stage, the redox condition was dysoxic, characterized by a higher lake level and more obvious reduction conditions than stage 2. There are two ways in which volcanic ash could have been brought into a closed lacustrine basin. The first is that volcanic ash fell on the land around the basin, and was carried into the water body by inflow water; the second is that volcanic eruption caused the volcanic ash to fall directly onto the surface of the water. Tuffaceous mudstone layers are not continuous in vertical distribution, and the paleo-salinity does not show an obviously corresponding fluctuation, which indicates that the alternated distribution of tuffaceous layers was mainly controlled by volcanic eruptions rather than by climate, despite a relatively small-scale

fluctuation in paleoclimate. Volcanic activity was frequent in the area near the western slope of Mahu Sag, and ash fell into the water with abundant nutrient elements, and tuff mudstones were developed, which enhanced the primary productivity. The relatively calm water conditions were favorable for the preservation of organic matter and the development of organic-rich mudstone. The vertical change in TOC shows that the organic matter at this stage was much more enriched than that at stage 2. However, at the early sedimentary stage of $P_1f_3$, the sedimentary rate was relatively high, which was not favorable for the preservation of organic matter. The comprehensive change in the paleo-environment indicates that the preservation of organic matter is not affected by any single controlling factor, but by the interaction of preservation and production conditions.

At the stage 4, namely the late depositional stage of $P_1f_3$, the water body was in overall dysoxic to oxic conditions. Due to the humid climate and high precipitation, the paleo-salinity of water became lower, with relatively frequent fluctuations in lake level. High lake level strengthened the connection between the sedimentary lake basin and the surrounding environment, providing it with adequate nutrient materials, and increasing the primary productivity and concentration degree of organic matter. Humid climate conditions were favorable for the reproduction of organisms, and biological activities were frequent. The decomposition of numerous organisms consumed a large amount of oxygen after death, forming a transient reduction condition and relatively high primary productivity. For sedimentary rate, stage 4 was lower than stage 3. But the humid climate, high oxidation conditions, and low water salinity were unfavorable conditions for the preservation of organic matter. Therefore, the primary controlling factor for the accumulation of the organic matter is the paleo-environment.

## 6. Conclusions

The enrichment of organic matter is controlled by many factors, not by any single factor. The absolute values of the partial correlation coefficient of the controlling factors increased in order from redox conditions, paleoclimate, terrigenous input, paleo-salinity, and paleo-productivity. The paleoclimate and paleo-productivity had a relatively weak negative influence on the accumulation of organic matter; the paleo-salinity, redox conditions, and detrital input had a weak positive influence on the concentration of organic matter. These relationships indicated that the accumulation of organic matter in the $P_1f$ was controlled by the combination of preservation and production conditions, rather than a single preservation condition or production condition.

For $P_1f$ in Well F7, three of the four stages of alkaline lake sedimentary evolution can be observed, including the early onset, preliminary, and weak terminal stages of alkalinity. Based on the paleoclimate, paleo-salinity, redox conditions, primary productivity, terrigenous influx, hydrothermal activity, and sedimentary rate, the depositional period of $P_1f$ can be divided into four stages. Stage 1 and stage 2 correspond to the early onset stage and preliminary stage, respectively, while stage 3 and stage 4 are in the weak terminal stage of alkalinity. The elemental indicator (Ti/Al) indicates that the detrital input was limited but fluctuating during the depositional period of $P_1f$. Additionally, there are two phases of hydrothermal activity at the early and middle-late stage of $P_1f_2$, respectively. At the late stage of $P_1f_1$, the paleo-environment of the water body was characterized by dysoxic conditions and an arid environment with relatively high salinity and relatively low paleo-productivity. At the early-middle stage of $P_1f_2$, the paleoclimate became more arid and the paleo-salinity increased, with a high sedimentary rate at the middle stage $P_1f_2$ and stronger reducibility, and the paleo-productivity was relatively low. From the late $P_1f_2$ to the middle stage of $P_1f_3$, the paleoclimate showed seasonally frequent fluctuations from semiarid to semi-humid, and generally became more warm and humid with dysoxic conditions while decreasing in salinity, as well as displaying moderate paleo-productivity. The high sedimentary rate at the early stage of $P_1f_3$ was not favorable for the preservation of organic matter. At the end of $P_1f_3$, the water body was overall in dysoxic to oxic conditions. Due to the humid climate and high precipitation, the water salinity was low.

**Supplementary Materials:** The following supporting information can be downloaded at: https:// www.mdpi.com/article/10.3390/pr11082483/s1, Table S1: Concentrations of REE and calculated parameters in samples from the Fengcheng formation in Mahu Sag. Table S2: Concentrations of major elements and TOC in samples from the Fengcheng formation in Mahu Sag. Table S3: Concentrations of trace elements in samples from the Fengcheng formation in Mahu Sag.

**Author Contributions:** Conceptualization, G.C.; methodology, G.C.; software, F.Y.; investigation, D.W.; writing—original draft preparation, Y.T.; writing—review and editing, Y.N. All authors have read and agreed to the published version of the manuscript.

**Funding:** This work was financed by the Karamay City Innovative Environment Building Program (Innovative Talents) of the Karamay Municipal Government, No. 20232023hjcxrc0036.

**Data Availability Statement:** The data supporting the reported results are displayed in the Supplementary Materials.

**Conflicts of Interest:** The authors declare no conflict of interest.

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
