# Peer review of "Paleo-Sedimentary Environments and Controlling Factors for Enrichment of Organic Matter in Alkaline Lake Sediments: A Case Study of the Lower Permian Fengcheng Formation in Well F7 at the Western Slope of Mahu Sag, Junggar Basin"

_processes, doi:10.3390/pr11082483_

Round 1
Reviewer 1 Report
Dear Editor,
I carefully read the manuscript of Chen et al. In general, the work is interesting and well illustrated. I think that it can be published in the journal after some corrections.
Row 13: You should clarify what P1f means. Of course, one can guess that this means the
Lower Permian Fengcheng formation, but this should be stated directly.
Row 39: It would be interesting to see several sentences that would describe the arguments in favor of the fact that sediments accumulated in lacustrine environments. There is a reference here to a paper about this, but still I would like to see briefly the arguments confirming the lake conditions.
Figure 1: In the Legend for the map, you should explain in more detail what Sag is, what Upper is, etc.
Row 173: It's not very clear what "asterisk" means. Is it a multiplication sign? Maybe there should be a "+" sign instead of the "*" sign? (A similar "asterisk" stands next to Ce*. It is clear that this means a normalization option, but, in this case, instead of the "multiply" sign in the form "*", it is better to put it in the form "x" or another).
Table 1.: I believe that the names of the rocks should be added here.
Table 2: Why didn't you mention the name of the rock?
Row 209: You actually used oxides, not elements, right?
Row 277: Could you briefly explain why the ratios of Sr/Cu are very sensitive to the change of the paleoclimate?
Rows 315-316: Barite crystallization often characterizes hydrothermal conditions, have you considered this in any way?
Rows 326-327: Y/Ho values of modern seawater varied between 44 and 74. Lower values are often associated with the influx of terrigenous or volcanic material.
Author Response
Please see the pdf file

Reviewer 2 Report
The abstract must be shortend by 50 %. Some further comments in the attached pdf.
Review Geochim
English grammar and the use of certain scientific vocabular needs a thorough improvement. For instance, the frequently used word “enrichment”, which describes a process, could be better replaced by “concentration” or “content”.
The samples are collected from cored sections, but the lithology is not mentioned. According to the litho-log, many samples are from the dolomitic sequences, but the CaO and MgO are relatively low. Volcanic input is mentioned but not shown in the litho profile.
How were the TOC-values analysed?
There are only five samples from the F1P1. According to the lithology four of them should be dolomite, but according to the high SiO2 concentration, more than 50 %, the classification as dolomite is problematic. The same inconsistencies can be found also with other samples.
The colour coding in Fig. 2 is not explained , who is the author of the profile?
Table 1: sum is not 100, missing LOI
The interpretation of the analyses uses in an irresponsible, unreflected way former results without a discussion of the necessary limits or errors. For instance, the use of element concentrations or element relations to estimate the humid/arid conditions in the catchment area should be limited to fine-grained muds, mudstones without a substantial carbonate content (Wei et al. 2020). However, the litho log is dominated by dolomitic rocks .
Many of the conclusions are highly speculative and not supported by the data. Trends are shown in several Figs. which are visible only for the authors. Blue sections are not defined.
Based on the geochemical data, the authors try to identify changes in the paleoclimatic conditions of the surrounding catchment area ,especially weathering and transport, the productivity of the alkaline water masses, volcanic input, the salinity of the water, even sea level changes and short rainfall can be detected. The authors claim that these processes are connected, but the uncritical arrangement of the indicators, frequently without a poor support by the data, cannot be accepted. For instance aolian input of a varying main terrigenous source is not discussed, the contribution of volcanic ashes will affect the geochemical analyses.
For some of the main problematic section see the comments in the pdf.revised. The comments are not completely covering the text, but should give some thoughts for the improvement of the ms.

see above
Reviewer 3 Report
Thank you for the opportunity to review this paper. It is a hard work, with a lot of data and you kind of get lost in them. That's why I would suggest, if the second table could be converted to ppm, because the others are in ppm and I know that the values can be converted and you would have uniformity, where is necessary. It can also be mentioned that ppm is the equivalent of mg/kg above teh table or in material and methods. I would suggest that the data from paragraphs 313-320 - 323, be in a table for a better visualization of the data. Its making you dizzy. And the same in the case of 358-365 .
Author Response
Dear Reviewer,
Thank you for your comments concerning our manuscript entitled “processes-2498177”. Your comments are all valuable and very helpful for revising and improving our paper, as well as the important guiding significance to our researches. We have studied comments carefully and have made correction which we hope meet with approval. Revised portion are marked in red in the paper. The main corrections in the paper and the responds to the reviewer’s comments are as following:
Responds to the reviewer’s comments:
1.Response to comment: if the second table could be converted to ppm, because the others are in ppm and I know that the values can be converted and you would have uniformity, where is necessary.
Thank you for your comments about convert Table 2 to ppm, and the value could be converted. If concentrations of major elements converted to ppm, the value will too big to display, Many articles on elemental geochemistry use percentage units to show that the major elements(https://doi.org/10.1016/j.jseaes.2022.105329).
2.Response to comment: I would suggest that the data from paragraphs 313-320 - 323, be in a table for a better visualization of the data. And the same in the case of 358-365.
Thank you for this suggestion for an article. We added two tables within the article to indicate the data, marked in red on the paper.
Mr. Yuhang Nan
Corresponding Author
Round 2
Reviewer 1 Report
Dear editor, the authors have corrected the manuscript and now, in my opinion, it can be published in your journal.
Author Response
Thank you for your suggestions for our article.
Reviewer 3 Report
Thank you for take in consideration my comments and sugestions
Author Response
Thank you for your suggestions for our article, We have studied comments carefully and have made correction which we hope meet with approval.